# Zα and Zβ Localize ADAR1 to Flipons That Modulate Innate Immunity, Alternative Splicing, and Nonsynonymous RNA Editing

**DOI:** 10.3390/ijms26062422

**Published:** 2025-03-07

**Authors:** Alan Herbert, Oleksandr Cherednichenko, Terry P. Lybrand, Martin Egli, Maria Poptsova

**Affiliations:** 1Discovery, InsideOutBio, Charlestown, MA 02129, USA; 2International Laboratory of Bioinformatics, HSE University, 101000 Moscow, Russia; oleksandr.cherednichenko@umu.se (O.C.); mpoptsova@hse.ru (M.P.); 3Department of Chemistry, School of Medicine, Vanderbilt University, Nashville, TN 37232-0146, USA; terry.p.lybrand@vanderbilt.edu; 4Center for Structural Biology, School of Medicine, Vanderbilt University, Nashville, TN 37232-0146, USA; 5Department of Biochemistry, School of Medicine, Vanderbilt University, Nashville, TN 37232-0146, USA; martin.egli@vanderbilt.edu

**Keywords:** ADAR1, flipons, G-quadruplex, Z-RNA, RNA editing, RNA splicing, R-loops, ALU repeats

## Abstract

The double-stranded RNA editing enzyme ADAR1 connects two forms of genetic programming, one based on codons and the other on flipons. ADAR1 recodes codons in pre-mRNA by deaminating adenosine to form inosine, which is translated as guanosine. ADAR1 also plays essential roles in the immune defense against viruses and cancers by recognizing left-handed Z-DNA and Z-RNA (collectively called ZNA). Here, we review various aspects of ADAR1 biology, starting with codons and progressing to flipons. ADAR1 has two major isoforms, with the p110 protein lacking the p150 Zα domain that binds ZNAs with high affinity. The p150 isoform is induced by interferon and targets ALU inverted repeats, a class of endogenous retroelement that promotes their transcription and retrotransposition by incorporating Z-flipons that encode ZNAs and G-flipons that form G-quadruplexes (GQ). Both p150 and p110 include the Zβ domain that is related to Zα but does not bind ZNAs. Here we report strong evidence that Zβ binds the GQ that are formed co-transcriptionally by ALU repeats and within R-loops. By binding GQ, ADAR1 suppresses ALU-mediated alternative splicing, generates most of the reported nonsynonymous edits and promotes R-loop resolution. The recognition of the various alternative nucleic acid conformations by ADAR1 connects genetic programming by flipons with the encoding of information by codons. The findings suggest that incorporating G-flipons into editmers might improve the therapeutic editing efficacy of ADAR1.

## 1. Introduction

ADAR1 (adenosine deaminase RNA specific) is an enzyme that can edit double-stranded RNA (dsRNA) by deaminating adenosine to form inosine. Its actions prevent the induction of interferon responses (Figure 1), and generate nonsynonymous codon edits in some 2261 human genes [1]. ADAR1 has multiple domains, three of which enable the recognition of dsRNA [2]; another, called Zα, that binds to left-handed Z-DNA and Z-RNA (collectively called ZNA) [3] (Figure 1A); a less well characterized Zβ domain that does not bind Z-DNA; and a C-terminal deaminase domain that catalyzes the replacement of the adenosine N6 amino group with a keto group, yielding inosine (A-to-I edit) in double-stranded (dsRNA) editing substrates. Most edits in human cells are to dsRNA substrates formed by ALU inverted repeats (AIR) [4,5,6,7,8,9]. ALUs are a form of SINE (Short Interspersed Nuclear Elements), with over one million copies in the human genome with the highest density in the GC-rich isochores [10]. Other endogenous repeat elements (EREs), such as LINES (Long Interspersed Nuclear Elements) and LTRs (long terminal repeats), also undergo editing. The binding of ADAR1 to the Z-RNA formed by these transcripts terminates the interferon response against host-encoded RNAs [11,12,13,14,15,16,17].

ADAR editing can also produce nonsynonymous codon changes to proteins by co- or post-transcriptional editing of their transcripts. The nucleotide substitution recodes the proteins since inosine is translated as guanosine [18]. In such cases, editing yields protein isoforms that differ by a single amino acid [19]. ADAR1 has two major isoforms, with the p110 protein lacking the Zα domain present at the N-terminus of the p150 protein. By recognizing specific nucleic acid structures, ADAR1 can make sequence-specific changes. ADAR1 can also alter the isoforms produced from a gene by editing splice acceptor sites [20]. The changes are soft-wired and variable rather than being fixed as in the case of hard-wired codon variations, resulting in the production of multiple RNA isoforms from pre-mRNAs [21].

Here, we review recent discoveries related to ADAR1 biology, starting with codon editing and moving on to outcomes dependent on flipons, sequences that adopt alternative nucleic acid conformations under physiological conditions [22]. Besides flipons that form ZNA, other sequences can also adopt alternative conformations. Different flipon types can fold into three-stranded triplexes, a four-stranded guanosine-rich quadruplexes (GQ) or an i-motif by intercalating cytosine base pairs [23,24,25,26,27,28,29,30,31]. We review how the recognition of alternative nucleic acid conformations by ADAR1 connects genetic programming by flipons with the encoding of information by codons. We then describe a novel interaction of Zα and Zβ with GQ that helps explain some unresolved issues in ADAR1 biology. Specifically, we discuss how the overlap between p150 and p110 editing substrates arises, the disagreements concerning the frequency of non-synonymous RNA editing (NSE) [32,33,34], and the role of ADAR1 in alternative pre-mRNA splicing. We also propose that ADAR1 promotes resolution of the R-loops by binding to GQ that form when an RNA transcript hybridizes with one DNA strand and displaces the other [35].

## 2. ADAR1 Isoforms

The edits by ADAR1 vary with the localization of each isoform in the cell. The p110 isoform is expressed constitutively and is predominantly nuclear. Here, the enzyme can edit exons that form dsRNA substrates by base pairing with intronic exon-complementary sites (ECS) [36]. Nuclear pre-mRNA editing is also associated with alternative splicing and suppression of circular RNA formation [20,37,38,39]. These findings build on earlier observations that ALU elements contain splice site motifs [40,41,42]. Indeed, 5% of alternatively spliced human exons are Alu-derived, with over 80% of these splices causing frameshifts or premature termination codons [40]. ADAR1 can also edit microRNA precursors [43,44]. 

The interferon-induced p150 isoform shuttles between the nucleus and cytoplasm and predominantly edits cytoplasmic dsRNA substrates produced by viruses or in cells with dysregulated transcription. Both p110 and p150 isoforms edit an overlapping subset of dsRNAs [16,17,45,46,47]. The human genome encodes other ADAR family members. ADAR2 (encoded by *ADARB1*) lacks Zα and Zβ domains and ADAR3 (encoded by *ADARB2*) lacks deaminase activity [48]. 

## 3. Codons and Editing

The initial discovery of RNA editing raised the question of whether the recoding of mRNA contributed to phenotype. Indeed, there was much excitement when the glutamine to arginine (QR) editing of the GRIA2 glutamate receptor was found, as this substitution changed the calcium conductance of the receptor [19]. However, mice with knockouts of the ADAR2 gene that performs the glutamine to arginine ion channel edit of the GRIA2 glutamate receptor were phenotypically normal if the edit was hardwired into the genome, suggesting that this edit rescued an otherwise deleterious mutation [49].

However, knockout of the *Adar* gene in mice was embryonic lethal, with death due to an interferonopathy [50,51,52]. Interestingly, the *Adar* null mice were rescued by expression of ADAR1 p150 but not p110, suggesting that p150 played a specific role in regulating interferon responses [53]. In humans, ADAR1 loss-of-function deaminase variants were also found to induce Aicardi–Goutières type 6 interferonopathy (AGS6) [54]. Interestingly, the pairing of a loss-of-function Zα allele with a null allele also induced AGS6, even though the deaminase domain was fully functional in the loss-of-function allele [11,55]. These Zα dependent outcomes have been successfully modeled in mice [16,17,56]. In humans, the pairing of a null Zα allele with the wildtype gene instead produces Dyschromatosis Symmetrica Hereditaria that is marked by mixed hyper- and hypopigmented macules on the dorsal aspect of the hands and feet and freckle-like macules on the face, but with no long-term health issues reported [11,57].

An interferonopathy was also induced in mice with a genomically encoded, catalytically dead ADAR1 mutation. The mice were rescued to live birth by preventing the interferon response through deletion of the dsRNA sensor MDA5 protein (melanoma differentiation antigen 5 encoded by *Ifih1*). The mice were phenotypically normal, demonstrating that ADAR1 was not essential for body plan specification [51,52,58]. Subsequently, a small set of ADAR-dependent NSE has been described, with the dependence of each event on ADAR1 and ADAR2 characterized, both in mice [44,46,58] and in humans [1,33,59]. The exact number of NSE due to ADAR1 editing is the subject of some dispute, with some authors proposing that the recoding of exons is rare in humans [32,34]. In contrast, others note that editing events are highly variable across tissues, with many conserved between species [33]. Of the 2261 annotated human genes, NSE frequently recodes lysine to arginine, preventing ubiquitinylation of these sites [1]. By changing protein turnover, NSE alters phenotypes by disrupting the non-genetically templated scaffolds that direct cellular responses to perturbation [60].

Recent mouse studies have further investigated the impact of ADAR1 on interferon responses. A triple gene knockout of ADAR p150, MDA5, and PKR yielded mice that live to adulthood without any obvious neonatal or adult phenotypes [61]. A different triple gene knockout model of *Adar*, *Eisf2ak2*, and *Mavs* (encoding mitochondrial antiviral signaling protein through which MDA5 signals) also displays longevity [62]. These results confirm that RNA editing is not essential for normal development. Instead, ADAR1 primarily regulates dsRNA-mediated responses, PKR-dependent defenses, and cell death pathways all triggered by EREs, such as AIRs (Figure 1B). Such outcomes are described in more detail below.

## 4. EREs and Editing

The vast majority of ADAR1 edits are to AIRs, with millions documented in resources such as REDIPortal [59]. The SINEs involved have no coding function. Instead, they spread through the genome by a copy-and-paste mechanism, exapting flipons to promote their transcription and retrotransposition. ALUs were initially derived from the noncoding 7SL RNA of the signal recognition particle that transfers nascent peptides across membranes as they emerge from the ribosome. The invasion by SINEs depends on the capture by these small RNAs of the proteins encoded by other EREs. The LINE and LTR RNA-dependent RNA polymerases (RdRp) that they repurpose allow SINEs to copy and paste themselves into active genes throughout the genome. The ALU family of SINE monomers duplicated themselves over time to form a dimer consisting of left and right arms, each of which subsequently diversified in sequence. The insertion of new ALUs is often near an existing ALU, often occurring in an inverted orientation. RNAs transcribed from these inverted repeats can fold back onto each other to form dsRNA editing substrates for ADAR1 [50,51,52,53,63]. Combating such invasive elements appeared to be a key early function of ADAR1, which traces back to *Capsaspora owczarzaki*, a unicellular organism that is likely the first eukaryotic progenitor [64,65,66].

Recognizing these threats by shape-specific domains like Zα is a key part of the defense. To understand this mechanism, it is first necessary to know more about the conformations that ADAR1 senses. The recognized structures are encoded by genetic elements called flipons, which are subject to natural selection and are heritable because they are built into the genome. These alternative nucleic acid structures dynamically regulate the localization of ADAR1 within cells and the subsequent nucleic-acid- and protein-dependent responses.

## 5. Flipons and Editing

Flipons encode information by their shape [22]. They can change conformation under physiological conditions to trigger outcomes specific to their structure. They are encoded within the repetitive genome of which EREs are a part. The different types of flipons are characterized by the repeat sequence that gives form to a particular fold. For example, alternating purine pyrimidine repeats can invert the base pairs to form left-handed Z-DNA [23,24], while runs of guanosine can hydrogen-bond with each other to form tetrads that stack into G-quadruplexes (GQ) [25,26]. Purine and pyrimidine repeats can add a third strand to generate triplexes [27,28]. Cytosine base pairs can also intercalate with each other to form i-motifs [29,30,31]. That said, the conformation that a flipon will adopt is not always clear-cut. For example, while d(TG)_n_ favors Z-DNA, r(UG)_n_ crystalizes as a left-handed GQ [67], while triplet repeats can also form Z-DNA [68]. There is also substantial overlap between ZNA- and GQ-forming sequences in promoter regions [69]. The high frequency of flipons in the genome renders them uninformative as B-DNA. However, in their alternative conformation, they flag active parts of the genome for the cellular engines to engage. The encoding is binary where flipons can only adopt one fold or another. The flip occurs without altering the underlying DNA sequence and does not require strand breakage. G-quadruplexes enable a rather complex biology due to their higher stability than Z-DNA, with many proteins involved in the cycling of G-flipons from one state to another (discussed in [70,71,72,73,74,75,76]). Z-flipons allow for rapid responses. G-flipons depend more on a lock and load mechanism, with GQ discharged only under specific conditions.

The energy required to initiate a flip to an alternative conformation can be generated by polymerases, helicases, and other sources of helical stress, such as stretching and twisting [77,78]. Enzymes like topoisomerases can relax the tension and restore the right-handed DNA helical conformation, while helicases can unwind dsRNA with free ends to produce single-stranded RNA (ssRNA) [79,80]. The flipon state can also be biased by editing the epigenetic markings on DNA, RNA, or chromatin, or by proteins and RNAs that dock to one conformation or another via sequence- or structure-specific complexes [81,82,83,84,85,86,87]. Usually, structure-specific proteins can recognize folds formed by RNA, DNA, or DRHs (DNA:RNA hybrids), although the affinity for each of these ligands may vary.

However, the differences between how DNAs, RNAs, and DRHs fold into an alternative conformation influence the biology associated with each nucleic acid type. For Z-RNA to form, RNAs transcribed from a pair of DNA flipons with complementary sequences must base-pair together. Positioning inverted repeat flipons close to each other favors dsRNA (A-RNA) formation. Further, bulges and mismatches in the dsRNA formed favor a flip of A-RNA to Z-RNA, as the pairing defects lower the energy cost of forming a right-handed/left-handed AZ-RNA junction relative to a BZ-DNA junction [88,89]. For G-flipons, an RNA-GQ (rGQ) can be encoded by the same sequence that specifies a DNA GQ (dGQ)-forming flipon. However, rGQs are rendered more stable than dGQ by the ribose 2′-OH group, with only two tetrads required to fold an rGQ compared to the three for a dGQ [90,91]. The folding of the two-stack rGQs studied was much slower than measured for either the three-stack rGQ or dGQ, which show similar kinetics, while data for hGQ (a hybrid GQ that incorporates both RNA and DNA strands) has not been reported [90,91]. There are also differences in the ion-binding preferences of rGQ compared to dGQ, with dGQ preferring potassium, while there is a better accommodation of sodium within the rGQ tetrad core [92]. At physiological ion concentrations, telomeric rGQ and dGQ can fold in ~60 msec [90]. These structures can be generated co-transcriptionally as the polymerase unwinds the double helix. During this process, the non-template DNA strand becomes single-stranded and can either fold into a dGQ or form a hGQ [93,94]. Interestingly, inosine can also substitute for guanosine in some positions of a GQ. This finding raises the possibility that ADAR1 editing can promote GQ formation by replacing adenosine with inosine to complete a tetrad [95,96].

## 6. Zα Targets ZNAs

ADAR1 is one of two proteins in the human genome known to bind ZNA in a structure-specific manner. The other protein, ZBP1 (Z-DNA-Binding Protein 1), has a pair of Zα domains (Figure 1B). Many of the ZNAs recognized by Zα are encoded by the repetitive genome [8,97]. Besides the simple repeats that characterize flipons, many EREs, such as ALU elements, can adopt multiple alternative RNA and DNA conformations and give rise to different families of small RNAs [98,99,100]. The ALU elements, in particular, were an existential threat to the survival of early hominids as they were inserted into active genes and caused DNA damage wherever they landed. Initially, flipons were used by the host to target the dsRNA produced by SINEs and other EREs. Sequence-specific strategies based on microRNAs and proteins to limit ERE spread evolved later. Similarly, flipons underwent selection to further the host’s survival and the production of progeny.

Interestingly, flipons encoded by EREs now act as tags that define self. In this role, the ALU Z-Boxes recognized by Zα prevent the induction of interferon responses by localizing ADAR1 p150 to self-transcripts. The interaction squelches ZNA- and dsRNA-dependent immune responses [78,85]. The supporting evidence derives from many disciplines, including detailed studies by structural, physical, and biological chemists, the functional mapping of pathways by cellular biologists, and the genetic validation of these findings in both human and rodent studies [9,11,16,17,101,102,103,104,105,106,107,108,109,110,111,112,113,114,115,116].

The use of Z-boxes to inhibit responses against self-RNAs does not prevent the induction of anti-viral responses. The flipon-based mechanism is not something easily mimicked by pathogens. Whether or not Z-RNA forms depends on the length of the dsRNA, with the stochastic engagement of MDA5 at different sites seemingly creating sufficient tension in the intervening region to initiate the flip from A-RNA (helical pitch 24.6 Å) to the more extended Z-RNA helix (pitch 45.6 Å) [78]. Z-RNA formation localizes ADAR p150 and helicases like DHX9 that then edit and unwind the dsRNA [117]. Once the invader, ALU elements now enable self/non-self discrimination. They allow the cell to tell friend from foe (Figure 1B) [78].

Localizing ADAR1 to dsRNA also prevents the activation of ZBP1 and PKR. ADAR1 p150 competes with ZBP1 for binding to ZNA (Figure 1B). Otherwise, ZBP1, whose expression is greatly increased by interferon, triggers a regulated cell death pathway that leads to necroptosis. If not resolved by ADAR1, the Z-Boxes present in AIRs [8,9] (Figure 2B) and other EREs activate ZBP1. Such an interaction between ADAR1 and ZBP1 is now well supported by many different genetic studies (Figure 1B) [12,13,14,15]. 

ADAR1 binding to dsRNA also prevents the PKR-induced shutdown of translation. Following activation by dsRNA, PKR disengages transcripts with a 7-methylguanosine cap from ribosomes [118]. These transcripts accumulate in stress granules and form tangles by cross-hybridizing with complementary repeat sequences present in other RNAs. Sufficient tension can be generated within these snarls to induce the flip of A-RNA to Z-RNA [119,120]. Through localizing ADAR1 to Z-RNAs, the Zα domain facilitates the resolution of these tangles by enhancing the editing of the dsRNA formed, their degradation by inosine-specific nucleases, and their unraveling by ADAR1-associated helicases.

## 7. Zα and Zβ Both Target G-Quadruplexes

Interestingly, Zα is reported to interact with GQ in addition to binding Z-DNA. The interaction was experimentally demonstrated using binding assays, circular dichroism spectra, and NMR chemical shift perturbation analysis. The docking with GQ involved different residues from those that bind Z-DNA [121]. No subsequent reports or structure determinations have been published. Meanwhile, the role of Zβ that is shared by both p150 and p110 has remained a curiosity. The substitution in Zβ of the Z-DNA-specific tyrosine by isoleucine (bolded in Figure 2B) abrogates high-affinity binding to ZNA [104,122,123]. Whether Zβ binds to GQ has not been experimentally addressed. However, an AlphaFold3 model [124] for the binding of Zβ to GQ supports the possibility (Figure 2C). The structure has been refined using Molecular Dynamics Simulation (MDS) and is stable for over 1.5 ms. The helix α3 R328 residue of Zβ (bolded in Figure 2B) makes a crucial contact. Zα also docks similarly to both rGQ and dGQ, a result consistent with the earlier NMR study [125].

This interaction of Zβ with GQ is supported by a previous study that reported the enrichment of both rGQ and A-to-I edits in introns close to RNA splice sites [126]. The edits are likely due to p110, which is localized in the nucleus. The p110 isoform also engages R-loops that enable folding of the telomeric G-rich repeats into dGQ and rGQ [35,127,128]. The docking of ADAR1 to the R-loop would be favored by binding Zβ to GQ. More generally, the binding of ADAR1 isoforms to GQ in the nucleus would explain their colocalization and the overlap of p110 and p150 editing sites. The Zα and Zβ domains would engage the p150, while Zβ would position p110 (Figure 2). Within the cytoplasm, the Zα domain enables a different mechanism to localize ADAR1 p150 to the Z-RNAs formed by AIRs and other EREs to modulate innate immune responses rather than to direct the splicing or the recoding of pre-mRNAs [61,62,119,120,129,130] (Figure 1B).

The ALUs targeted by p110 and p150 editing have the potential to form GQs [85], with the sequences underscored in Figure 2D [4,5,7,131]. They are more likely to form rGQ than dGQ, as the 2′-OH renders a GQ formed from a stack of two RNA tetrads much more stable than one made from two DNA tetrads [90,91]. The different nuclear and cytoplasmic roles of Zα and Zβ domains possibly underlie the evolution of different ALU families. One set of ALUs contains a perfect ZNA-forming alternating cytosine and guanosine motif (Figure 1D), while other families have a GQ motif at this location (Figure 2D,K).

Overall, it is probable that ADAR1 isoforms localize to nascent transcripts through the interaction of Zβ with GQs. In this scenario, the initial docking of ADAR1 is to a hGQ formed near the site of transcription, most commonly to ALU-derived sequences (Figure 2D,E). The binding of ADAR1 fixes the 5′ end of the transcript as the RNA polymerase continues moving down the helix. While the RNA may be held in place through formation of a hGQ [92,93], it is possible that the rGQ stacks on the dGQ (Figure 2L,M). Modeling with Alphafold3 supports a stacking interaction, with Zβ binding preferentially to rGQ rather than to dGQ. MDS shows that the stacking is very stable, with eight water molecules forming a bridged hydrogen bonding network between the protein and RNA with favorable van der Waals contacts made by isoleucine and guanine bases. The tension generated by fixing the RNA end stretches the nascent transcript. The exposed exocyclic hydrogen-bonding base residues can then pair with upstream sequences to form dsRNA. ADAR1 then transfers to those with a dsRNA stem of at least 15 base pairs to engage the deaminase domain (Figure 2F) [132]. Other proteins that form complexes with ADAR1, such as DHX9 and DDX21, also help resolve the R-loops that led to GQ formation. Subsequently, proteins necessary for the splicing and processing of pre-miRNA and other small noncoding RNA fragments are loaded onto the nascent transcript [39,133,134,135].

In this process, ADAR1 may help ameliorate aberrant splices. The nuisance splices can arise from anti-sense ALU family members that cluster three 3′ AG splice acceptor sites adjacent to a polyuridine tract that promotes splicing (Figure 2J) [136]. All three splice sites change the reading frame of the isoform produced, with one splice option creating a UGA stop codon. The different outcomes lead to either the inclusion of an exon, loss of an exon, or termination of a reading frame. A GQ motif is immediately adjacent to these splice sites in the ALU-S family that has over 55,000 copies in the human genome [10]. In these cases, occlusion of the potential splice sites by the docking of ADAR1 to GQ through either Zα or Zβ, will inhibit aberrant splicing and ensure that the transcripts undergo triage rather than producing dysfunctional proteins that lack key domains or fold incorrectly when these defective RNAs undergo translation.

In some cases, the transcripts that form GQ will also form dsRNA editing substrates. We present several examples that exemplify several different scenarios (Figure 3). An AIR can produce long dsRNA with multiple edits and GQ motifs (Figure 3A–C). The intriguing possibility exists that the initial editing of these elements fosters the formation of additional GQs that are stabilized by the substitution of inosine for adenosine. The de novo GQs then promote the docking of multiple ADAR1 enzymes to the substrate leading to the hyper-editing of the transcript [95,96,137] that disrupts any splice sites in the region [138]. Figure 3D,E show that the well-known K242R edit of NEIL1 (endonuclease VIII-like DNA glycosylase) is associated with a GQ motif ([139,140,141,142]. With the TNFRSF14 transcript, A-to-I edits are associated with a GQ motif and alternative splicing events (Figure 3E,F) [1,32,33,34,143]. The edited FLG (filaggrin) RNA is also associated with both a GQ and an antisense CCDST (cervical cancer associated DHX9 suppressive transcript) RNA (Figure 3H,I). The sense, anti-sense pairing could potentially form a dsRNA editing substrate, but edits are only found on the sense strand, suggesting that folding of the FLG RNA on itself is sufficient to induce A-to-I substitutions (Figure 3I).

## 8. Zβ and NSE

A recent publication lists 2261 genes with NSE [34]. Our analysis revealed that 75% of the edited sequences were either at the beginning (17%) or end (59%) of a transcript, suggesting that they were recently added to the gene body. Only a few NSE instances intersect with an ALU sequence tag (160 genes containing 681 exons with edits overlapping an ALU fragment; overlap range = 5–322, median overlap = 122 bases, mean overlap = 162 bases). We further found that there were 661 genes (29.2%) in which editing was associated with alternative splicing, as noted above for ALU-associated edits. The findings suggest that the variability of NSE previously reported may reflect different alternative splicing frequencies within the cells and tissues studied by each group.

The efficiency of the editing events is also variable between cells [32,33,34]. Consequently, different transcripts arise from a single gene that increases the genetic variability on which natural selection is based [144]. Some novel variants will produce cells more adaptive to the current contingencies than their genomic twins and undergo selection to populate tissues. In some situations, selection favors the retention of an intronic ECS to compensate for an otherwise detrimental variant, as is the case for the QR editing of the GRIA2 glutamate receptor transcript [49]. Over time, the exonization of ALU elements and other EREs with subsequent optimization of their coding sequence will lock in particular phenotypes by hardwiring those particular variations into the genome. Flipons then become codons.

## 9. ZNAs, EREs, and Disease

The binding of Zα to Z-flipons has received much attention due to its association with disease [8]. ADAR1 and ZBP1 play essential roles in infections by bacteria and viruses, while EREs contribute to diseases through ZNA-dependent pathways. When unchecked by ADAR1, the ZBP1 and PKR pathways lead to inflammatory diseases triggered initially by either EREs or viral infections [13,97,145,146,147,148,149,150,151,152,153,154,155,156,157,158,159,160,161,162,163,164,165]. Roles for ZBP1 activation in autoimmunity have also been demonstrated in mouse models [160,166,167]. Interestingly, reverse transcriptase inhibitors prevent the association of ZBP1 with Z-DNA/Z-RNA hybrids in the stress granules of patients with systemic lupus erythematosus [168]. In contrast, ADAR1 expressed by tumors can induce immune silencing by suppressing the activation of interferon responses [169]. In a genome-wide CRISPR screen, many cancer cell lines derived from various tissues depended on *ADAR* for survival [170]. Notably, in a preclinical melanoma model, a CRISPR guide specific for the p150 isoform improved the response to checkpoint inhibitors [171]. In the same model, an intercalating agent that bypasses ADAR1 inhibition of ZBP1 to directly activate Z-DNA-dependent tumor cell death also improved the response to a checkpoint inhibitor [12]. Further, the induction of ERE expression with demethylating agents like decitabine to activate ZBP1 also improved anti-tumor responses [172]. Other strategies that interfere with removing EREs by splicing or triage are also under investigation [173]. A small molecule that selectively disrupts the production of the ADAR1 p150 led to leukemia stem cell death and was effective in preclinical models [174].

Attention has recently been directed to the many ways viruses have thwarted the ZNA-dependent innate immune response directed against them. One strategy is to incorporate the relatively small Zα domain into their genome. This strategy appears to have been first deployed by the ancient giant viruses that infected unicellular eukaryotes, as seen by both sequence homology and confirmed experimentally [66,175]. Modern-day viruses, such as pox and asfi viruses, also have Zα domains that act as virulence factors [114,176]. Other viruses employ various strategies to inhibit responses, such as the disruption they cause to normal host transcript termination. Unfortunately for the virus, this outcome leads to the enhanced expression of EREs lying in wait just outside the bounds of normal gene expression [177,178]. The ZNAs formed by these EREs then activate the host ZNA-dependent responses that protect the cell regardless of what induced the perturbation. Cancers and other aged cells past their expiry date also promote their own demise through the genome-wide promiscuous transcription that they promulgate, springing the trap set to prevent their persistence. Here, the ‘junk’ has undergone selection because the flipon-containing EREs help inform and protect the host. By monitoring the conformation of these flipons, the cell introspectively assesses its health status and responds accordingly.

The role of Zβ in disease has not been addressed, although NSE, AAS, and miRNA editing have been noted as risk factors in cancer by many authors, as well as a cause of cardiovascular disease [179,180,181,182,183]. The use of synthetic RNA to create the dsRNA substrates necessary for ADAR1 to recode transcripts in vivo is now a cutting-edge therapeutic endeavor [184,185]. The findings presented here raise the possibility that the therapeutic editing efficacy of ADAR1 might be improved by incorporating GQs into editmers.

## 10. Method Supplement

Figure 1A was generated using NGL Viewer [186].

Molecular dynamics Simulations were performed using the PMEMD AMBER module [187]. The AMBER OL21 force field [188] was used for DNA with the AMBER ff19SB force field for protein [189], along with the OPC water model [190] and corresponding parameters for monovalent ions [191]. The ADAR1–DNA complex was solvated in a truncated octahedron box with a 14 Å buffer zone between any complex atom and the closest box wall. The starting structure was subjected to a three-step minimization procedure. First, the protein–DNA complex was relaxed for 10,000 steps of conjugate gradient minimization while the water molecules and counterions were restrained at starting positions. Next, all solvent and counterions were relaxed for 10,000 steps while the complex was restrained. Finally, all restraints were removed, and the entire system was minimized over 10,000 additional steps. The minimized complex was then heated gradually from 0 to 300 K during a 100 ns canonical ensemble (NVT) MD simulation, followed by a ~200 ns NPT ensemble simulation, using a 2 fs timestep. Energy and force calculations were performed using minimal image periodic boundary conditions, a 12 Å nonbonded cutoff for real space interactions, a homogeneity assumption to approximate the contributions of long-range Lennard–Jones forces to the virial tensor, and staggered particle-mesh Ewald for long-range electrostatics correction [192]. A Langevin thermostat with a collision frequency of 3 ps^−1^ was used to maintain the system temperature [193]. All bonds containing hydrogen were constrained using the SHAKE algorithm [194] and the SETTLE method was used to maintain rigid water geometry [195]. Final numerical analysis of all MD trajectories was performed using the cpptraj package [196] and the ChimeraX 1.9 program was used for graphical analysis and the generation of Figure 2C [197]. The PDB file is given in the Appendix A.NRE from Li et al. [1] that results from A→I editing were mapped to HG19. A pattern search for GQ on the same strand as the edited adenosine was performed in a region +/−200 bp around NSE, using the following string: “G(2+N(1-10)G(2+)N(1-10) G(2+)N(1-10) G(2+)” (where “N” is any nucleotide, with the loop sizes specified by the numbers in brackets, and “+” indicating greater than or equal to the number preceding it) [90,198]. The results are given in Appendix A. Then, we searched for isoforms where the edited exons associated with GQ motifs overlapped the introns of other isoforms. For this purpose, we used transcript annotation taken from UCSC (Gencode v47 lift 37). The selected list of alternatively spliced exons was overlapped with ALU (RepeatMasker annotations). The results are given in Appendix A. Files for both the HG19 and HG38 genomes are provided.

## Figures and Tables

**Figure 1 ijms-26-02422-f001:**
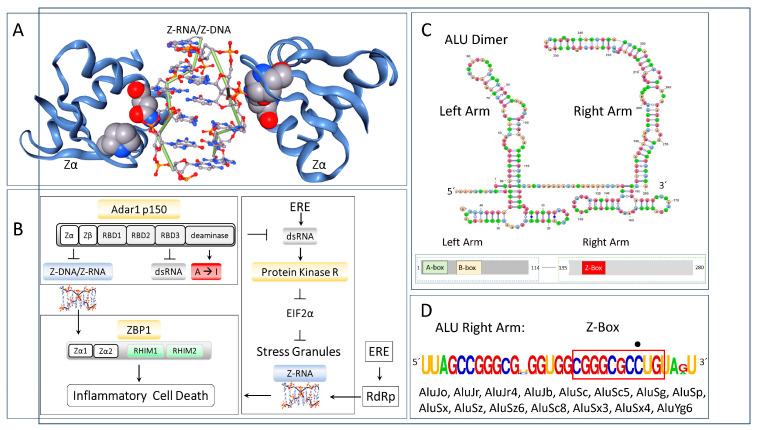
Zα and RNA editing in innate immunity. (**A**) Zα binds in a structure-specific fashion to Z-DNA with high affinity. The space-filling representations of P193 and N173 highlight their interaction with ZNA. Loss of function P193A and N175S variants are causal for the Aicardi–Goutières type 6 interferonopathy. (**B**) The only other protein in the human genome with a Zα domain is ZBP1. It activates inflammatory cell death in response to Z-RNA produced by viruses and endogenous retroelements (EREs) that amplify themselves by RdRp. Z-RNA also forms in stress granules. ADAR1 suppresses the Z-RNA-dependent activation of ZBP1. ADAR1 also prevents the dsRNA activation of PKR (protein kinase R encoded by EIF2AKA). PKR inhibits EIF2α (Eukaryotic Initiation Factor 2 alpha) dependent translation, leading to stress granule formation. ZBP1 interfaces with effector pathways through RHIMs (receptor-interacting protein homotypic interaction motif). (**C**) Most ALU elements are dimeric with a left and right arm. A- and B-boxes enable transcription by RNA Polymerase 3. The Z-Box present in ALU, as indicated by the red outline in (**D**), has one guanosine to cytosine substitution disrupting the expected pyrimidine/purine alternation (as indicated by the dot above the residue), The sequence is colored coded by base. The flip of the ALU Z-Box to Z-RNA helps identify transcripts as of host origin as pathogens lack this class of Z-forming element.

**Figure 2 ijms-26-02422-f002:**
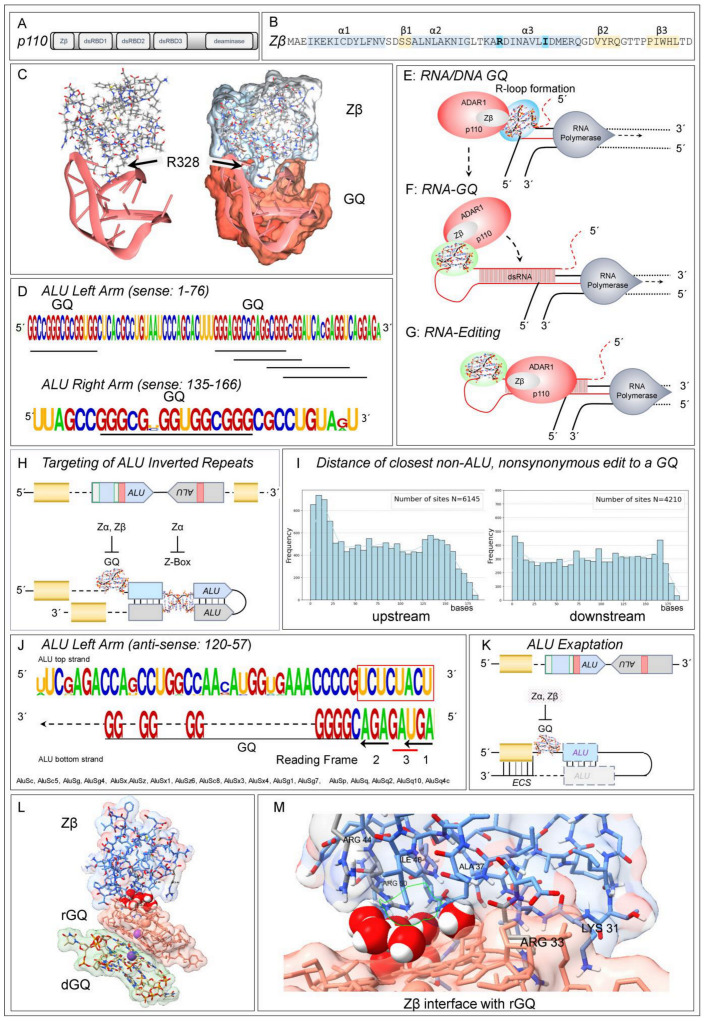
Zβ and RNA Editing. (**A**) ADAR1 p110 isoform domains. (**B**) Schematic of the Zβ winged helix-turn-helix domain, with the α-helices highlighted in blue and the β-sheet strands in yellow. The Y→I substitution that greatly diminishes binding to Z-DNA relative to Zα and the location of R328 in the α3 helix are shown with bold type face. (**C**) The docking of Zβ to a parallel strand GQ with U4 loops, modeled by Molecular Dynamics Simulation. R328 makes key contacts between the protein and DNA/RNA. (**D**) The sets of G-tetrad pairs that can form RNA-GQ are underlined. The right and left arms correspond to those in Figure 1C. (**E**) A model of how Zβ enables the loading of p110 first onto a GQ formed during transcription, then onto a folded dsRNA (**F**) to enable engagement of the deaminase domain (**G**). (**H**) ADAR1 p110 and p150 are localized to ALU inverted repeats by different flipon conformations. (**I**) The sites of NSE are not associated with ALU elements but rather with GQs that are in close proximity. (**J**) GQ motifs on the antisense strand of some ALU families are close to a cluster of 3′ “AG” splice acceptor sites. Two of these “AG” sites change the reading frame and the third introduces a UAG stop codon into a reading frame. The coding potential of isoforms is further varied by Zβ-dependent editing of the “AG” splice sites and exonic NSE. (**K**) Through selection, ALU exonization, alternative splicing, and NSE can increase phenotypic diversity. The GQ and ALU elements may be incorporated into an exon or downstream of a splice junction. In the latter case, the GQ promotes editing of the exon when the downstream exon contains an exon complementary site (ECS), while the remnants of an ALU inverted repeat promote folding of the dsRNA editing substrate. In other cases, an ALU-derived GQ may be associated with the 3′ splice junction. Editing of the “AG” acceptor site would result in exon skipping. (**L**) Molecular Dynamics Simulation of the ADAR1 Zβ domain complexed with rGQ (pink surface) stacked on dGQ (green surface). (**M**) Close-up view of the interface between Zβ and rGQ. The Arginine 33 side chain penetrates deeply into the rGQ molecule, forming multiple hydrogen bonds and favorable van der Waals contacts with three guanine residues and the RNA backbone. Lysine 31 (corresponding to K326 in human ADAR1 p150) forms hydrogen bonds and a favorable ion-pair interaction with the RNA backbone. Eight water molecules within the interface form hydrogen bonding networks that bridge the protein backbone and amino acid sidechains with the RNA backbones and bases.

**Figure 3 ijms-26-02422-f003:**
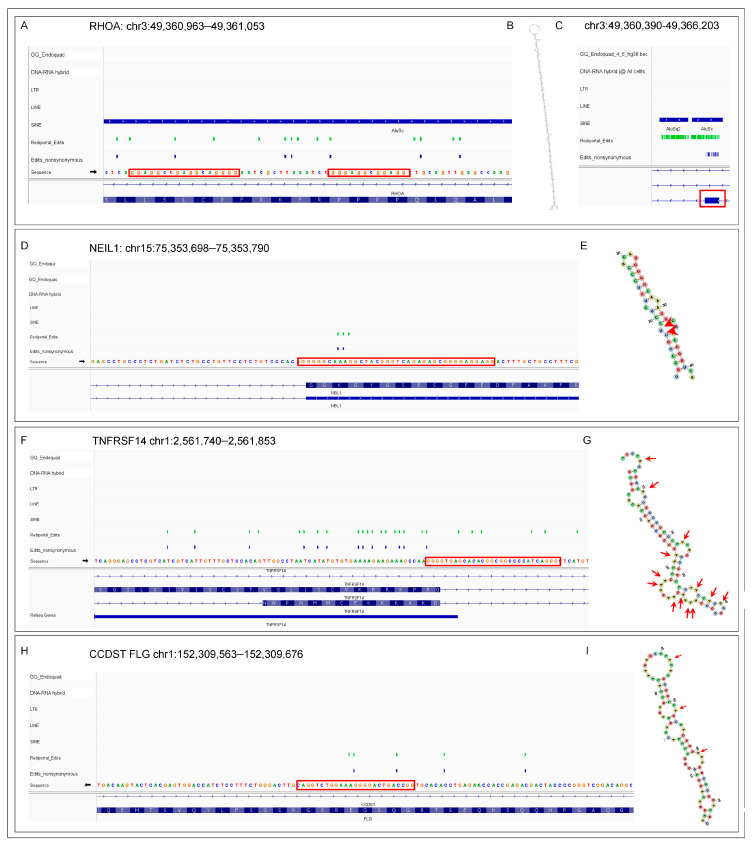
Examples of the GQ forming sequences associated with RNA editing. (**A**) ALU-associated editing of the RHOA transcript. The red boxes highlight the GQ. (**B**) The long dsRNA editing substrate formed by the fold back of the two ALU inverted repeats displayed in (**C**). The red box shows the position of the RHOA exon relative to the ALU elements. (**D**) The well-characterized K242R edit of NEIL1 (endonuclease VIII-like DNA glycosylase) is associated with a GQ. (**E**) The NEIL1 editing substrate with the edits indicated by red arrows. (**F**) The edits and various RNA isoforms of TNFRSF14 pre-mRNA are associated with a GQ motif. (**G**) Location of the edits in a dsRNA substrate formed by the TNFRSF14 transcript. (**H**) The FLG (filaggrin) RNA is associated with both a GQ and the antisense CCDST (cervical cancer associated DHX9 suppressive transcript) that could form a dsRNA substrate. (**I**) The FLG pre-mRNA by itself also folds into a dsRNA substrate.

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
