# Peer review of "Zα and Zβ Localize ADAR1 to Flipons That Modulate Innate Immunity, Alternative Splicing, and Nonsynonymous RNA Editing"

_ijms, 2025, doi:10.3390/ijms26062422_

Round 1

Reviewer 1 Report

Comments and Suggestions for Authors

This paper offers a systematic review of recent advancements in ADAR1 biology, with a particular focus on the roles of the Zα and Zβ domains in nucleic acid structure recognition and their influence on gene programming diversity. It also explores potential optimization strategies for therapeutic editing. While the topic holds substantial scientific importance, the manuscript requires improvements in logical coherence, terminology standardization, and linguistic precision.

Comments:

  1. Spelling Errors: Multiple spelling errors need correction (e.g., "essential" → "essential", "recognition" → "recognition", "specific" → "specific"). Additionally, some instances of "ffipons" appear to be misspelled as "flipons"; the terminology should be standardized throughout the text. Grammar Issues: There are grammatical inconsistencies, such as the subject-verb disagreement in "increase phenotypic diversity" (should be "increases phenotypic diversity"). The phrase "moving on the ffipons" contains a prepositional error and should be revised to "moving to flipons."
  2. The introduction should more clearly outline the unresolved key questions in ADAR1 research. For instance, the molecular basis of the functional distinctions between Zα and Zβ domains, as well as the mechanisms by which G-quadruplex (GQ) structures specifically recruit ADAR1, should be explicitly addressed.
  3. The manuscript mentions "strong evidence" but fails to specify the nature of this evidence (e.g., structural studies, loss-of-function experiments, or genomic analyses). It is essential to provide references to key data sources or cite relevant studies to substantiate these claims.
  4. The conclusion that incorporating GQ structures into editmers can enhance editing efficiency lacks detailed mechanistic justification or experimental validation. It is recommended to include a theoretical framework or cite prior research to support this proposition.
  5. The term "flipons" should be clearly defined in the introduction, along with its relationship to retrotransposons, to prevent conceptual ambiguity and ensure reader comprehension.
  6. The manuscript should discuss whether ADAR1-mediated RNA editing directly drives alternative splicing or if these processes occur independently. Citing mechanistic studies, such as editing site mutation experiments, would strengthen the argument and provide clarity on this relationship.
Comments on the Quality of English Language

The English could be improved to more clearly express the research. 

Author Response

Thanks for the time you spent going through the paper so extensively. The paper has been extensively rewritten to address the many issues you raise.

Comments:

  1. Spelling Errors: Multiple spelling errors need correction (e.g., "essential" → "essential", "recognition" → "recognition", "specific" → "specific"). Additionally, some instances of "ffipons" appear to be misspelled as "flipons"; the terminology should be standardized throughout the text. Grammar Issues: There are grammatical inconsistencies, such as the subject-verb disagreement in "increase phenotypic diversity" (should be "increases phenotypic diversity"). The phrase "moving on the ffipons" contains a prepositional error and should be revised to "moving to flipons."

These have been corrected using Grammarly as suggested by Reviewer 2

  1. The introduction should more clearly outline the unresolved key questions in ADAR1 research. For instance, the molecular basis of the functional distinctions between Zα and Zβ domains, as well as the mechanisms by which G-quadruplex (GQ) structures specifically recruit ADAR1, should be explicitly addressed.

The introduction has been rewritten to address these issues. On line 42, we write

“Here, we review recent discoveries related to ADAR1 biology, starting with codons and moving on to flipons, sequences that adopt alternative nucleic acid conformations under physiological conditions [22]. Besides flipons that form ZNA, other sequences that we describe later can fold into three-stranded triplexes and four-stranded guanosine-rich quadruplexes (GQ) or weave together like i-motifs [23-31]. We describe how the recognition by ADAR1 of alternative nucleic acid conformations connects genetic programming by flipons with that dependent upon codons. We then describe a novel interaction of Zα and Zβ with GQ that helps explain some unresolved issues in ADAR1 biology. Specifically, we discuss how the overlap between p150 and p110 editing substrates arises, the disagreements concerning the frequency of non-synonymous RNA editing (NSE) [32-34], the impact on alternative pre-mRNA splicing, and the role of ADAR1 in the resolution of the R-loops that form when an RNA displaces one DNA strand from the double helix by hybridizing with the other strand [35].”

  1. The manuscript mentions "strong evidence" but fails to specify the nature of this evidence (e.g., structural studies, loss-of-function experiments, or genomic analyses). It is essential to provide references to key data sources or cite relevant studies to substantiate these claims.

We are not sure which part of the paper this comment references. If it is the wording used in the abstract (line 10), we changed “strong evidence to “strong supporting evidence”. A description of the supporting evidence is the subject of the paper.

If the comments relate to the biological function of Zα, we reference such evidence extensively throughout the paper. An extensive  listing of 20 references is given on line 183 at the end of this sentence “The supporting evidence derives from many disciplines, including detailed studies by structural, physical, and biological chemists, the functional mapping of pathways by cellular biologists, and the genetic validation of these findings in both human and rodent studies [9, 11, 16, 17, 101-116].”

  1. The conclusion that incorporating GQ structures into editmers can enhance editing efficiency lacks detailed mechanistic justification or experimental validation. It is recommended to include a theoretical framework or cite prior research to support this proposition.

We did not make that conclusion, only the suggestion, both as the last line of the abstract and of the paper

“The findings suggest that the therapeutic editing efficacy of ADAR1 might be improved by incorporating GQs into editmers.” (lines 14 and 326)

We include a thorough theoretical discussion based on experimental support. Figure 2 incorporates data from ALphafold3 and Molecular Dynamics simulations, as indicated in the supplemental methods.

  1. The term "flipons" should be clearly defined in the introduction and its relationship to retrotransposons to prevent conceptual ambiguity and ensure reader comprehension.

We modified the introduction to give a working definition of flipons,  noting that a fuller description will come later, starting on line 40

Here, we review recent discoveries related to ADAR1 biology, starting with codons and moving on to flipons, sequences that adopt alternative nucleic acid conformations under physiological conditions [22]. Besides flipons that form ZNA, other sequences that we describe later can fold into three-stranded triplexes and four-stranded guanosine-rich quadruplexes (GQ) or weave together like i-motifs [23-31]. We describe how the recognition by ADAR1 of alternative nucleic acid conformations connects genetic programming by flipons with that dependent upon codons. We then describe a novel interaction of Zα and Zβ with GQ that helps explain some unresolved issues in ADAR1 biology. Specifically, we discuss how the overlap between p150 and p110 editing substrates arises, the disagreements concerning the frequency of non-synonymous RNA editing (NSE) [32-34], the impact on alternative pre-mRNA splicing, and the role of ADAR1 in the resolution of the R-loops that form when an RNA displaces one DNA strand from the double helix by hybridizing with the other strand [35].”

We also have a new section entitled “ERE and Editing” (line 106)

The full discussion is in the section titled  “Editing and Flipons”, (line 126.)

  1. The manuscript should discuss whether ADAR1-mediated RNA editing directly drives alternative splicing or if these processes occur independently. Citing mechanistic studies, such as editing site mutation experiments, would strengthen the argument and clarify this relationship.

We have now referred to these papers in the introduction starting  line 56

“The edits by ADAR1 vary with the localization of each isoform in the cell. The p110 isoform is expressed constitutively and is predominantly nuclear. Here, the enzyme can edit exons that form by the foldback of introns containing exon-complementary sites (ECS) [36]. Nuclear pre-mRNA editing is also associated with alternative splicing and suppressing circular RNA formation [20, 37-39]. The findings build on earlier observations that ALU elements contain splice site motifs [40-42]. Indeed, 5% of human alternatively spliced exons are Alu-derived, and most ALU-containing exons are alternatively spliced. Over 80% of the ALU-associated splices (AAS) cause a frameshift or premature termination codon [40]. “

The paper addresses mechanisms extensively in many places.

Reviewer 2 Report

Comments and Suggestions for Authors

This manuscript, "Zα and Zβ localize ADAR1 to flipons that modulate innate immunity, alternative splicing, and nonsynonymous RNA editing", reviews the mechanism of action of the double-stranded RNA editing enzyme ADAR1 and its key role in regulating gene expression and immune response. This manuscript firstly describes the basic principles of non-synonymous mutation editing by ADAR1 through recognition of specific nucleotide sequences, followed by a detailed discussion of the functional differences between the two main isoforms of ADAR1 (p110 and p150), where p150 is able to bind to dsRNAs and inhibit the interferon response upon interferon induction, and p110, which functions mainly in the nucleus, by recognising and editing specific repetitive sequences for gene editing. The role of nucleotide sequences in the editing process is also specifically emphasised. This paper reveals in detail the importance of ADAR1 in the development of disease and informs the enzyme's role in maintaining genetic stability and regulating biodiversity. The article should be revised for the following issues before accept, and it has fulfilled the requirements of the journal submission.

Line 54-58More examples of immune disorders caused by Adar gene deletion should be provided, as well as more examples of p150-p110 inter-regulation.

Line 90-92The critical early function of ADAR1 should be more described and presented in multiple species.

Line 111-112Information about things that are well known to the general public can be less illuminating, such as the many sources of energy required for conformational flipping.

Line 163-166Research from many disciplines, including physical and biochemists, cell biologists on the functional localization of pathways requires evidence of specific research content.

Line 290Financial support for this manuscript needs to be provided.

Comments on the Quality of English Language

none

Author Response

Thanks for your time in reviewing the paper and for the helpful suggestions you provided.

Line 54-58:More examples of immune disorders caused by Adar gene deletion should be provided, as well as more examples of p150-p110 inter-regulation.

We have moved all such discussion to the last section staring line 290

“ZNAs, EREs, and Disease

The binding of Zα to Z-flipons has received much attention due to its association with disease [8]. ADAR1 and ZBP1 play important roles in infections by bacteria and viruses, while EREs contribute to diseases through ZNA-dependent pathways. When unchecked by ADAR1, the ZBP1 and PKR pathways lead to inflammatory diseases that are triggered initially by either ERE or viral infections [13, 97, 145-165]. Roles for ZBP1 activation in autoimmunity have also been demonstrated in mouse models [160, 166, 167]. Interestingly, reverse transcriptase inhibitors prevent the association of ZBP1 with Z-DNA/Z-RNA hybrids in stress granules of patients with systemic lupus erythematosus [168]. In contrast, ADAR1 expressed by tumors can induce immune silencing by suppressing the activation of interferon responses [169]. In a genome-wide CRISPR screen, many cancer cell lines derived from various tissues depended on ADAR for survival [170]. Notably, in a preclinical melanoma model, a CRISPR guide specific for the p150 isoform improved the response to checkpoint inhibitors [171]. In the same model, an intercalating agent that bypasses ADAR1 inhibition of ZBP1 to activate Z-DNA-dependent tumor cell death directly also improved the response to a checkpoint inhibitor [12]. Further, induction of ERE expression with demethylating agents like decitabine to activate ZBP1 also improved anti-tumor responses [172]. Other strategies that interfere with removing ERE by splicing or by triage are also under investigation [173]. A small molecule that selectively disrupts the production of the ADAR1 p150 leads to leukemia stem cell death and is effective in preclinical models [174].

Attention has recently been directed to the many ways viruses have thwarted the ZNA-dependent innate immune response directed at them. One strategy is to incorporate the relatively small Zα domain into their genome. This strategy appears to have been first deployed by the ancient giant viruses that infect unicellular eukaryotes, as seen by both sequence homology and confirmed experimentally [66, 175]. Modern-day viruses, such as pox and asfi viruses. also have Zα domains that act as virulence factors [114, 176]. Other viruses employ various strategies to inhibit responses, such as the disruption they cause of normal host transcript termination. Unfortunately for the virus, this outcome leads to the enhanced expression of EREs that lie in wait just outside the bounds of normal gene expression [177, 178]. The ZNAs formed by these EREs then activate the host ZNA-dependent responses that serve to protect the cell regardless of what induced the perturbation. Cancers and other aged cells past their expiry date also promote their own demise through the genome-wide promiscuous transcription that they promulgate, springing the trap that is set to prevent their persistence. Here, the ‘junk’ has undergone selection because the flipon containing EREs helps to both inform and protect the host. By monitoring the conformation of these flipons, the cell introspectively assesses its health status and responds accordingly.

The role of Zβ in disease has not been addressed, although NSE, AAS, and miRNA editing have been noted as risk factors in cancer by many authors, as well as cardiovascular disease [179-183], The use of synthetic RNA to create in vivo the dsRNA substrates necessary for ADAR1 to recode transcripts is now a cutting-edge therapeutic endeavor [184, 185]. The findings presented here raise the possibility that the therapeutic editing efficacy of ADAR1 might be improved by incorporating GQs into editmers.”

Line 90-92:The critical early function of ADAR1 should be more described and presented in multiple species.

We have moved all this information to  a section named Codon and Editing starting line 69

Codons and Editing

The initial discovery of RNA editing raised the question of whether the recoding of mRNA contributed to phenotype. Indeed, there was much excitement when the glutamine to arginine (QR) editing of the GRIA2 glutamate receptor was found as this substitution changed the calcium conductance of the receptor [19]. However, mice with knockout of the ADAR2 gene that perform the glutamine to arginine ion channel edit of the GRIA2 glutamate receptor were phenotypically normal if the edit was hardwired into the genome, suggesting that this edit rescued an otherwise deleterious mutation [49].

However, knockout of the Adar gene in mice was embryonic-lethal, with death due to an interferonopathy [50-52]. Interestingly, the Adar null mice were rescued by expression of p150 but not p110, suggesting that p150 played a specific role in regulating interferon responses [53]. In humans, ADAR1 loss-of-function deaminase variants were also found to induce the Aicardi Goutières Type 6 interferonopathy (AGS6) [54]. Interestingly, the pairing of a loss-of-function Zα allele with a null allele also induced AGS6, even though the deaminase domain was fully functional in the loss-of-function allele [11, 55]. These outcomes have been successfully modeled in mice [16, 17, 56]. In humans, the pairing of a null Zα allele with the wildtype gene produces Dyschromatosis Symmetrica Hereditaria that is marked by mixed hyper- and hypopigmented macules on the dorsal aspect of the hands and feet and freckle-like macules on the face, but with no long term health issues reported [11, 57].

An interferonopathy was also induced in mice with a genomically encoded, catalytically dead ADAR1 mutation. The mice were rescued to live birth by preventing the interferon response by deleting the dsRNA sensor MDA5 protein (melanoma differentiation antigen 5 encoded by Ifih1). The mice were phenotypically normal, demonstrating that ADAR1 was not essential for the specification of the body plan [51, 52, 58]. Subsequently, a small set of ADAR-dependent NSE has been described, with the dependence of each event on ADAR1 and ADAR2 characterized, both in mice [44, 46, 58] and in humans [1, 33, 59]. The exact number of NSE is the subject of some dispute, with some authors proposing that the recoding of exons is rare in humans [32, 34], while others note that the events are highly variable across tissues, with many conserved between species [33]. Of the 2261 annotated human genes, NSEs frequently recode lysine to arginine, preventing the ubiquitinylation of these sites [1]. Like other non-genetically templated scaffolds that regulate protein turnover, the NSEs impact cellular responses to perturbations [60].

Recent mouse studies have further investigated the impact of ADAR1 on interferon responses. A triple gene knockout of ADAR p150, MDA5, and PKR (Protein Kinase RNA, encoded by EIF2AK2 that is activated by dsRNA) yielded mice that lived to adulthood without any obvious neonatal or adult phenotype [61]. A different triple gene knockout model of Adar, Eisf2ak2, and Mavs (encodes mitochondrial antiviral signaling protein through which MDA5 signals also displayed longevity [62]. The results confirm that RNA editing is not essential for normal development. Instead, ADAR1 primarily regulates interferon responses to dsRNA, PKR-dependent defenses, and cell death pathways all triggered by ERE, such as AIRs (Figure 1B). These outcomes are described in more detail below.

Line 111-112:Information about things that are well known to the general public can be less illuminating, such as the many sources of energy required for conformational flipping.

In our experience, this information is not as well-known as you believe, nor is it known that the transitions occur without cleavage of the DNA backbone,

Line 163-166:Research from many disciplines, including physical and biochemists, cell biologists on the functional localization of pathways requires evidence of specific research content.

Much of this work concerns Zα. This paper is about Zβ.

We do give an extensive list of references detailing the experimental work relating to Zα. Key elements, such as the structure of Zα bound to Z-DNA  and ALU RNA, are given in Figure 1, along with the pathways discussed and referenced elsewhere in the paper. The dsRNA fold-back structures and models proposed in the paper are given in Figure 2, and examples of A-to-I edited RNAs are presented in Figure 3. There are many recent reviews written about Zα domains.

Line 290:Financial support for this manuscript needs to be provided.

There was no grant support for this work.

Reviewer 3 Report

Comments and Suggestions for Authors

This review article covers the recognition of two alternative nucleic acid structures, the Z-form of DNA and RNA and the G-quadraplexes in RNA, by three domains of the ADAR enzyme. These alternative structures are part of larger flipons within nucleic acids that induce the flip to the alternative conformation under certain physiological conditions. The lead author is the world-leading expert on the topic of Z-DNA binding proteins and has been working on it for four decades.

The lead author publishes review articles on this general them on a regular basis so the question is whether or not this review makes a unique contribution. The review is unqiue in showing how the ADAR binding can link the formation of Z-flipons and G-flipons.  This binding is turn bridges RNA editing and RNA splicing. This is an exciting idea that will be of great interest to the readers of IJMS.

The Introduction could be improved by adding a paragraph about flipons, which are a relatively new concept that is not widely known.

The authors do a good job of defining the acronyms that they utilize, but they fail to define the acronym ADAR1. Some readers attracted to this topic may not remember the definition of this acronym. 

Regarding the issue of self-citation, there is a moderately high number of self-citations. However, I think that many of these are justified because the lead author has played a seminal role in this field for three decades. The lead author needs to remove some non-essential citations of his review artilces that he has written in the past. The citations of his research papers are valid because of the vital role of these papers in this field.

The literature-cited section needs to be cleaned up. Many of the citations list only the first page of the article and not the range of pages. Some journal names have incorrect capitalization.
For example, the name of the journal `RNA' is given incorrectly as `Rna'. These errors likely come from incomplete or inaccurate entries in the bibliographic database.

Comments on the Quality of English Language

Regarding the quality of the written English, there are several cases of inappropriate use of commas, and there were several wrong verb tenses. Some sentences had complex structures that were difficult to follow. These sentences could be split into two or more; these simplifications would help the reader by improving the clarity.
The authors would benefit from running the text through an application like Grammarly to check the punctuation and verb tenses.

Author Response

The lead author is the world-leading expert on the topic of Z-DNA binding proteins and has been working on it for four decades.

It is nice to see that these alternative structures now have experimental validation of their biological functions.

The lead author publishes review articles on this general them on a regular basis so the question is whether or not this review makes a unique contribution. The review is unqiue in showing how the ADAR binding can link the formation of Z-flipons and G-flipons.  This binding is turn bridges RNA editing and RNA splicing. This is an exciting idea that will be of great interest to the readers of IJMS.

Thanks for your kind comments.

Since this is a new field, these “reviews” synthesize pre-existing data and new analyses to help others understand the impact on their work in various fields. While generally about flipons, the subject matter in each paper is very different.

The Introduction could be improved by adding a paragraph about flipons, which are a relatively new concept that is not widely known.

Thanks for the suggestion. We expanded the introduction to include the definition of flipons and to better describe the paper's contents, starting on line 40.

“Here, we review recent discoveries related to ADAR1 biology, starting with codons and moving on to flipons, sequences that adopt alternative nucleic acid conformations under physiological conditions [22]. Besides flipons that form ZNA, other sequences that we describe later can fold into three-stranded triplexes and four-stranded guanosine-rich quadruplexes (GQ) or weave together like i-motifs [23-31]. We describe how the recognition by ADAR1 of alternative nucleic acid conformations connects genetic programming by flipons with that dependent upon codons. We then describe a novel interaction of Zα and Zβ with GQ that helps explain some unresolved issues in ADAR1 biology. Specifically, we discuss how the overlap between p150 and p110 editing substrates arises, the disagreements concerning the frequency of non-synonymous RNA editing (NSE) [32-34], the impact on alternative pre-mRNA splicing, and the role of ADAR1 in the resolution of the R-loops that form when an RNA displaces one DNA strand from the double helix by hybridizing with the other strand [35]..”

The authors do a good job of defining the acronyms that they utilize, but they fail to define the acronym ADAR1. Some readers attracted to this topic may not remember the definition of this acronym.

We have described the acronym on the first line of the introduction

Regarding the issue of self-citation, there is a moderately high number of self-citations. However, I think that many of these are justified because the lead author has played a seminal role in this field for three decades. The lead author needs to remove some non-essential citations of his review artilces that he has written in the past. The citations of his research papers are valid because of the vital role of these papers in this field.

Thanks for describing the situation so clearly. However,  I have only worked in the field for 15 years, with a 15-year gap when it was not possible for me to do so (no one thought that the work was worth funding). The fact that nothing much happened to advance the field during the gap is quite remarkable.

The literature-cited section needs to be cleaned up. Many of the citations list only the first page of the article and not the range of pages. Some journal names have incorrect capitalization.

For example, the name of the journal `RNA' is given incorrectly as `Rna'. These errors likely come from incomplete or inaccurate entries in the bibliographic database.

The problem was with EndNote and is corrected – some e-journals only have a single epage for an article, not a page range

Comments on the Quality of English Language

Regarding the quality of the written English, there are several cases of inappropriate use of commas, and there were several wrong verb tenses. Some sentences had complex structures that were difficult to follow. These sentences could be split into two or more; these simplifications would help the reader by improving the clarity.

The authors would benefit from running the text through an application like Grammarly to check the punctuation and verb tenses.

Thanks for the suggestion!

Round 2

Reviewer 1 Report

Comments and Suggestions for Authors

The authors have addressed all the comments. The improved manuscript can be acceptable for publication in the Journal. Thank you. 

Reviewer 2 Report

Comments and Suggestions for Authors

no further comment